# An Improved Elephant Herding Optimization for Energy-Saving Assembly Job Shop Scheduling Problem with Transportation Times

**Tianhua Jiang** [1,2,3,*] , **Lu Liu** [1,3] , **Huiqi Zhu** [1] and **Yaping Li** [1]

1   School of Transportation, Ludong University, Yantai 264025, China
2   Key Laboratory of Symbolic Computation and Knowledge Engineering of Ministry of Education, Changchun 130012, China
3   Shandong Marine Aerospace Equipment Technological Innovation Center, Ludong University, Yantai 264025, China
*   Correspondence: jth1127@163.com

**Abstract:** The energy-saving scheduling problem (ESSP) has gained increasing attention of researchers in the manufacturing field. However, there is a lack of studies on ESSPs in the assembly job shop environment. In contrast with traditional scheduling problems, the assembly job shop scheduling problem (AJSP) adds the additional consideration of hierarchical precedence constraints between different jobs of each final product. This paper focuses on developing a methodology for an energy-saving assembly job shop scheduling problem with job transportation times. Firstly, a mathematical model is constructed with the objective of minimizing total energy consumption. Secondly, an improved elephant herding optimization (IEHO) is proposed by considering the problem's characteristics. Finally, thirty-two different instances are designed to verify the performance of the proposed algorithm. Computational results and statistical data demonstrate that the IEHO has advantages over other algorithms in terms of the solving accuracy for the considered problem.

**Keywords:** energy-saving scheduling; assembly job shop; total energy consumption; improved elephant herding optimization

**MSC:** 97P10

## 1. Introduction

According to the relevant survey, the worldwide industrial sector will consume more than half of the total energy through 2040 [1]. Against the background of the deteriorating ecological environment, environmental protection has become a serious issue for global manufacturing industries. Both economic benefits and environmental factors compel manufacturing managers to introduce some promising techniques to control energy consumption. In recent years, energy-saving scheduling has become a new research direction in the manufacturing field [2]. In this problem, an optimal scheduling scheme is drawn up to reduce energy consumption from the production management perspective. With a slight extra financial burden on enterprises, the energy-saving scheduling attracts more attention in comparison with some traditional methods, such as purchasing energy-efficient equipment and designing new products, especially for some small-scale enterprises [3].

In recent decades, the flexible job shop scheduling problem (FJSP) has attracted great attention from researchers due to its wide application and high complexity [4]. Along with the promotion of green manufacturing, the energy-saving FJSP has gradually become a research hotspot in recent years [5–13]. Like the classical FJSP, the previous work on the energy-saving FJSP assumes that operations in each job of the workshop must be processed following a predefined sequential precedence constraint, while jobs are independent from

each other. However, for real-life complex products in many machining-assembly manufacturing systems, such as automobile engines, bicycles, etc., they are made up of multiple and multilevel jobs and characterized by Bills-Of-Materials (BOMs). This means that not only do sequential operation precedence constraints exist in each job, but hierarchical precedence constraints between different jobs also need to be considered simultaneously [14]. In general, the variant of FJSP with hierarchical job precedence constraints is named the assembly job shop scheduling problem (AJSP). In the existing literature, AJSP can be classified into two types: two-stage AJSP and hybrid AJSP. In the two-stage AJSP, the machining and assembly operations are handled in the machining and assembly stages, respectively. The traditional FJSP is contained in the machining stage, which deals with the sequential precedence relationships between operations in each job. The assembly stage is attached to the machining stage to carry out the assembly operations once all jobs of the same BOM are completed or available after the machining stage [15–20]. This means that the line-structure operation precedence constraints and the tree-structure job precedence constraints are considered separately [14], which leads to the splitting of the inherent parallel relationship between machining and assembly operations. Compared with the two-stage AJSP, the hybrid AJSP mixes machining operations and assembly operations. The mixture of hierarchical and sequential precedence constraints significantly increases the complexity and makes the hybrid AJSP problem quite challenging to solve. Zhu and Zhou [14] proposed a multi-objective grey wolf optimization algorithm to solve the FJSP with job precedence constraints to minimize the makespan, the maximum machine workload, and the total machine workload simultaneously. Pathumnakul and Egbelu [21] considered the AJSP problem in just-in-time manufacturing with the objective of optimizing the weighted earliness cost. A heuristic algorithm was developed by decomposing the problem into several single machine scheduling problems. Chen et al. [22] studied a FJSP with hierarchical job precedence constraints. An actual weapons manufacturing factory was used as a case study to test the performance of the proposed dispatching rules. Na and Park [23] considered a FJSP with multi-level job structures. A hybrid genetic algorithm was proposed to minimize the total tardiness of jobs. Jiang and Wang [24] studied the AJSP in an aircraft-engine assembly workshop and proposed a genetic algorithm to solve the problem. Zou et al. [25] proposed a level-based evolutionary algorithm to deal with the AJSP.

In the previous work about AJSP, researchers only concentrated on improving production efficiency, such as makespan, earliness, tardiness, workload, etc. In the current context of green manufacturing, some ecological metrics should be addressed in the energy-saving AJSP to meet the needs of sustainable development, such as energy consumption, noise pollution, $CO_2$ emission, and carbon footprint. To the best of our knowledge, there is little literature focusing on the energy-saving AJSP. Ren et al. [26] developed a mathematical model with the objective of improving production efficiency and minimizing energy consumption. Then, a heuristic PSO-GA algorithm was proposed to solve the problem. However, the considered scheduling problem is a two-stage AJSP rather than a hybrid AJSP. In addition, the transportation operations of jobs between machines are neglected to simplify the problem. In fact, production and transportation operations strongly interact with each other. On one hand, the machine selection of two successive operations in a job determines the transportation time. On the other hand, the transportation time can affect the waiting times of machines in terms of different operation sequences [3]. Furthermore, a certain amount of energy will be generated in the transportation process. Thus, the job transportation times should also be considered to narrow the gap between the scheduling problem and the practical production.

Since the considered energy-saving AJSP is the extended version of the classical FJSP, the problem has the nature of NP-hard. It is well-known that it is difficult to obtain the optimal solutions to production scheduling problems using exact methods, even for small-scale problems. Therefore, intelligence algorithms are an effective alternative method due to the advantage of finding satisfactory solutions in an affordable time. In recent years, various intelligence algorithms have been successfully applied for solving the energy-

saving scheduling problems, such as African buffalo optimization (ABO) [2], genetic algorithm (GA) [3,5,6], interior search algorithm (ISA) [7], cat swarm optimization (CSO) [8], imperialist competitive algorithm (ICA) [9], pigeon-inspired optimization (PIO) [10], bat algorithm (BA) [11], particle swarm optimization (PSO) [26], etc. However, the No Free Lunch (NFL) theorem [27] implies that no algorithm performs best for all optimization problems, which means that a particular intelligence algorithm may attain promising solutions on a set of problems, but it may show poor performance on a different set of problems. This motivates researchers to explore new meta-heuristics for coping with various optimization problems. Thus, it is worthwhile to develop a fresh optimization algorithm for the energy-saving assembly job shop scheduling problem.

Inspired by the social behaviors of elephants in nature, elephant herding optimization (EHO) a novel swarm intelligence algorithm. Due to the impressive advantages of EHO, including easy implementation and good convergence, it has been successfully implemented for various optimization problems [28–31]. The primary difference with most of the existing meta-heuristics is that EHO employs a multi-population technique in the evolutionary process. In this technique, the elephant population is split into several sub-populations, based on which differentiated search strategies can be easily adopted among sub-populations and the collaboration and communication between sub-populations can be easily implemented. Therefore, the disadvantages of the premature convergence and the loss of population diversity can be well overcome, which are suffered by most of the existing meta-heuristic algorithms. This is the main reason to choose the EHO for solving the considered problem. Furthermore, the basic EHO algorithm was originally presented for continuous optimization problems. Thus, some problem-oriented modifications need to be carried out to make it suitable for the discrete scheduling problem. These reasons motivate the systematic investigation of the EHO for the considered energy-saving assembly job shop scheduling problem. The main works of this paper are summarized as follows: (1) A mathematical model is established for the energy-saving assembly job shop scheduling problem with transportation times; (2) a two-segment string is adopted to represent scheduling solutions, and an energy-saving decoding method is proposed to obtain active scheduling solutions; (3) a population initialization is utilized to obtain initial solutions with a certain quality and diversity; (4) two searching operators, namely the clan updating operator and separating operator, are designed to implement the improved elephant herding optimization algorithm (IEHO) following the characteristics of the problem.

The remainder of this paper is structured in the following manner. Section 2 formulates the considered scheduling problem. Section 3 shows the proposed IEHO algorithm in detail, including the encoding/decoding approach, population initialization, and discrete search operators. Section 4 conducts extensive experiments to verify the performance of the proposed algorithm. Section 5 reports the conclusion and future work.

## 2. Problem Description and Mathematical Model

### 2.1. Problem Description

The energy-saving AJSP can be described as follows: In a workshop, there are $n$ products $\{P_1, P_2, \cdots, P_n\}$ to be manufactured by $m$ machines $\{M_1, M_2, \cdots, M_m\}$. Each product is composed of $I_i$ jobs $\{JB_{i1}, JB_{i2}, \cdots, JB_{iI_i}\}$ corresponding to a tree-structure BOM. Each job is consisted by $J_{ij}$ operations $\{O_{ij1}, O_{ij2}, \cdots, O_{ijJ_{ij}}\}$ with a line-structure precedence relationship. For each operation $O_{ijq}$, it must be processed on a machine chosen from its eligible machine set. The processing time of $O_{ij}$ depends on the processing capacity of the selected machine. In addition, an operation consumes a different amount of energy when it is assigned to different machines. When a job is finished on a machine, it will be immediately conveyed to the next machine for machining or assembly. The transportation times between different machines are assumed to be known. The transfer also consumes an amount of energy during the transportation time. This problem attempts to assign operations to an appropriate machine and sequence them on each machine. The objective is to optimize the total energy consumption, which is composed by four types:

processing energy consumption (*PEC*), idle energy consumption (*IEC*), transportation energy consumption (*TEC*), and auxiliary energy consumption (*AEC*). Some assumptions are considered as follows:

(1) All jobs are released and all machines are available at time zero.
(2) Each machine can perform one operation simultaneously.
(3) Each job can only be processed by one machine at a time.
(4) The processing of each operation cannot be interrupted.
(5) Each machine cannot be turned off until all jobs on it are completed.
(6) There are enough transfers for transportation operations between machines.
(7) The last operation of each product does not need to be transported after it is finished.
(8) Machine breakdown is negligible and setup times of machines are ignored.

### 2.2. Mathematical Model

Before describing the problem, some necessary symbols are shown as below.

$i$: The index of products, $i = 1, 2, 3, \cdots, n$;
$j$: The index of jobs, $j = 1, 2, 3, \cdots, I_i$;
$q$: The index of operations, $q = 1, 2, 3, \cdots, J_{ij}$;
$k$: The index of machines, $k = 1, 2, 3, \cdots, m$;
$O_{ijq}$: The $q$th operation of job $j$ in product $i$;
$O_{\mathrm{P}(ijq)}$: The immediate successor operation of $O_{ijq}$;
$p_{ijqk}$: The processing time of $O_{ijq}$ when it is processed on machine $k$;
$F_1$: The objective function;
$TOE$: Total energy consumption;
$PE_{ijqk}$: Processing energy consumption coefficient of $O_{ijq}$ when it is processed on machine $k$;
$IE_k$: Idle energy consumption coefficient of machine $k$ when it is idle;
$AE$: Auxiliary energy consumption coefficient;
$TE$: Transportation energy consumption coefficient;
$C_k$: Completion time of machine $k$;
$S_k$: Start time of machine $k$;
$WL_k$: Workload of machine $k$, the sum of the processing times of jobs on machine $k$;
$C_{\max}$: The final completion time (makespan);
$TT_{ijqw,\mathrm{P}(ijq)k}$: The transportation time between machine $w$ and machine $k$ for $O_{ijq}$ and $O_{\mathrm{P}(ijq)}$;
$ST_{ijq}$: Starting time of $O_{ijq}$;
$CT_{ij}$: Completion time of $O_{ijq}$;
$\Gamma$: A positive number big enough for Constraints (8) and (9);
$x_{ijqk}$: A binary variable, if $O_{ijq}$ is processed on machine $k$, $x_{ijqk} = 1$; otherwise, $x_{ijqk} = 0$;
$y_{ijqi'j'q'k}$: A binary variable, if $O_{ijq}$ is processed before $O_{i'j'q'}$ adjacently on machine $k$, $y_{ijqi'j'q'k} = 1$; otherwise, $y_{ijqi'j'q'k} = 0$.

$$F_1 = \min TOE = \min(PEC + IEC + TEC + AEC) \tag{1}$$

$$\mathrm{s.t.} PEC = \sum_{i=1}^{n} \sum_{j=1}^{I_i} \sum_{q=1}^{J_{ij}} \sum_{k=1}^{m} PE_{ijqk} p_{ijqk} x_{ijqk} \tag{2}$$

$$IEC = \sum_{k=1}^{m} IE_k (C_k - S_k - WL_k) \tag{3}$$

$$TEC = \sum_{i=1}^{n} \sum_{j=1}^{I_i} \sum_{q=1}^{J_{ij}} \sum_{w=1}^{m} \sum_{k=1}^{m} TE \cdot TT_{ijqw,\mathrm{P}(ijq)k} x_{ijw} x_{\mathrm{P}(ijq)k} \tag{4}$$

$$AEC = AE \times C_{\max} \tag{5}$$

$$CT_{ijq} - ST_{ijq} = \sum_{k=1}^{m} x_{ijqk} p_{ijqk}, \quad i = 1, 2, \cdots, n; \ j = 1, 2, \cdots, I_i; \ q = 1, 2, \cdots, J_{ij} \tag{6}$$

$$ST_{P(ijq)} \geq CT_{ijq} + \sum_{w=1}^{m} \sum_{k=1}^{m} TT_{ijqw,P(ijq)k} x_{ijqw} x_{P(ijq)k}, \quad i = 1, 2, \cdots, n; \ j = 1, 2, \cdots, I_i; \ q = 1, 2, \cdots, J_{ij} \tag{7}$$

$$ST_{i'j'q'} + \Gamma(1 - y_{ijqi'j'q'k}) \geq CT_{ijq'},$$
$$i, i' = 1, 2, \cdots, n; j(j') = 1, 2, \cdots, I_i(I_{i'}); q(q') = 1, 2, \cdots, J_{ij}(J_{i'j'}); k = 1, 2, \cdots, m \tag{8}$$

$$ST_{ijq} + \Gamma y_{ijqi'j'q'k} \geq CT_{i'j'q'}, \quad i, i' = 1, 2, \cdots, n; j(j') = 1, 2, \cdots, I_i(I_{i'}); q(q') = 1, 2, \cdots, J_{ij}(J_{i'j'}); k = 1, 2, \cdots, m \tag{9}$$

$$\sum_{k=1}^{m} x_{ijqk} = 1, \quad i = 1, 2, \cdots, n; \ j = 1, 2, \cdots, I_i; \ q = 1, 2, \cdots, J_{ij} \tag{10}$$

$$WL_k = \sum_{i=1}^{n} \sum_{j=1}^{I_i} \sum_{q=1}^{J_{ij}} p_{ijqk} x_{ijqk}, \quad k = 1, 2, \cdots, m \tag{11}$$

$$C_k = \max\left\{ CT_{ijq} x_{ijqk} \right\}, \quad i = 1, 2, \cdots, n; \ j = 1, 2, \cdots, I_i; \ q = 1, 2, \cdots, J_{ij}; \ k = 1, 2, \cdots, m \tag{12}$$

$$S_k = \min\left\{ ST_{ijq} x_{ijqk} \right\}, \quad i = 1, 2, \cdots, n; \ j = 1, 2, \cdots, I_i; \ q = 1, 2, \cdots, J_{ij}; \ k = 1, 2, \cdots, m \tag{13}$$

$$ST_{ijq} \geq 0, \quad i = 1, 2, \cdots, n; \ j = 1, 2, \cdots, I_i; \ q = 1, 2, \cdots, J_{ij} \tag{14}$$

$$x_{ijqk} \in \{0, 1\}, \quad i = 1, 2, \cdots, n; \ j = 1, 2, \cdots, I_i; \ q = 1, 2, \cdots, J_{ij}; \ k = 1, 2, \cdots, m \tag{15}$$

$$y_{ijqi'j'q'k} \in \{0, 1\}, \quad i, i' = 1, 2, \cdots, n; j(j') = 1, 2, \cdots, I_i(I_{i'}); q(q') = 1, 2, \cdots, J_{ij}(J_{i'j'}); k = 1, 2, \cdots, m \tag{16}$$

Equation (1) denotes the optimization objective of the problem. Constraint (2) denotes the processing energy consumption, which originates from the processing of operations on machines. Constraint (3) represents the idle energy consumed by machines when waiting for processing tasks. Constraint (4) gives the transportation energy consumption, which is generated by transfers for transporting jobs between machines. Constraint (5) defines the energy consumed by auxiliary equipment, such as lighting and air conditioning. Constraint (6) means that preemption is not allowed. Constraint (7) gives the precedence relationships between operations. Constraints (8) and (9) guarantee that each machine cannot process more than one operation at a time. When $y_{ijqi'j'q'k} = 1$, $O_{ijq}$ is processed before $O_{i'j'q'}$ on machine $k$. Accordingly, Constraint (8) holds. $\Gamma$ is set to ensure that Constraint (9) also holds. When $y_{ijqi'j'q'k} = 0$, $O_{ijq}$ is processed behind $O_{i'j'q'}$ on machine $k$. Accordingly, Constraint (9) holds. $\Gamma$ is used to let Constraint (8) also be met; Constraint (10) ensures that any operation can only choose one machine for its processing; Constraint (11) calculates the machine workload; Constraints (12) and (13) give the completion time and starting time of each machine; Constraint (14) represents that the start time of each operation is greater than zero; Constraints (15) and (16) state two 0–1 variables.

## 3. Overview of the Basic EHO Algorithm

The elephant herding optimization (EHO) algorithm was proposed by Wang et al. [28] according to the herding behavior of elephants. In the EHO, two operators are used to formulate the herding behavior of elephants, i.e., the clan updating operator and separating operator. A brief introduction of the basic EHO algorithm is presented as follows.

### 3.1. Clan Updating Operator

In each clan, each elephant updates its position under the guidance of the fittest elephant corresponding to the natural matriarch. For the elephant $i$ in clan $c$, the updating operation can be formulated by Equation (17).

$$X_{ci}^{t+1} = X_{ci}^{t} + \alpha \times (X_{c,best}^{t} - X_{ci}^{t}) \times r \tag{17}$$

where $t$ is the current generation; $X_{ci}$ is the position of individual $i$ in clan $c$; $\alpha \in [0,1]$ is a scale factor which reflects the influence of the matriarch on $X_{ci}$; $X_{c,best}^{t}$ represents the matriarch's position in clan $c$. $r$ is a random number with the uniform distribution in $[0,1]$. In addition, for the matriarch, its position is updated by Equation (18).

$$X_{ci}^{t+1} = \beta \times X_{c,center}^{t} \tag{18}$$

$$X_{c,center,d}^{t} = \frac{1}{n_c} \times \sum_{i=1}^{n_c} X_{c,i,d}^{t} \tag{19}$$

where $\beta$ is a scale factor in $[0,1]$; $X_{c,center}^{t}$ is the center of the clan, which can be calculated by Equation (19); $n_c$ is the number of elephants in clan $c$; $X_{c,center,d}^{t}$ and $X_{c,i,d}^{t}$ are the $d$th dimension of $X_{c,center}^{t}$ and $X_{ci}^{t}$, respectively.

### 3.2. Separating Operator

The separating operator simulates the leaving of the mature male elephants from their groups. This operator is conducted to the individual elephants with the worst fitness value, as shown in Equation (20).

$$X_{c,worst}^{t+1} = X_{\min} + (X_{\max} - X_{\min} + 1) \times rand \tag{20}$$

where $X_{\min}$ and $X_{\max}$ are the lower and upper bounds of the elephant's position. $X_{c,worst}$ represents the worst individual in clan $c$. $rand$ is a random number with a uniform distribution in $[0,1]$.

### 3.3. Elitism Strategy

Like some other intelligence algorithms, the elitism strategy is adopted to avoid the best individuals from being ruined during the evolutionary process. In the beginning of each generation, the *nKEL* best elephant individuals are saved, which are used to replace the *nKEL* worst elephant individuals at the end of every search process. This strategy guarantees that the later elephant population is not worse than the former one.

### 3.4. Steps of the Basic EHO

The detailed steps of the basic EHO algorithm are shown as below.

Step 1. Create the initial population at random, and determine the parameters of the algorithm, such as the population size *popsize*, the number of saved best elephants *nKEL*, the maximum generation *maxiter*, the scale factors $\alpha$ and $\beta$, the number of clans *nclan*, and the number of elephants $n_c$ in the $c$th clan.

Step 2. Evaluate the fitness of each elephant.

Step 3. Sort all the elephants in the population according to the fitness values, save the *nKEL* best elephants, and then divide the population into *nclan* clans.

Step 4. Conduct the clan updating operator following Equations (17) and (18).

Step 5. Conduct the separating operator following Equation (20).

Step 6. Combine the clans into one population, and evaluate the fitness of all new individuals.

Step 7. Conduct the elitism strategy to replace the worst individuals with the *nKEL* saved ones.

Step 8. Judge whether the maximum generation is met. If yes, go to Step 9, otherwise, go to Step 3.

Step 9. Output the results.

## 4. Implementation of the IEHO

### 4.1. Encoding Approach

To implement the IEHO, the first step is to devise an encoding approach to represent the solution to the problem. Here, a two-segment string is applied to express each schedul-

ing solution. The first segment stores the information of machine assignment (MA) for each operation, and the second gives the information of operation permutation (OP) on each assigned machine. The length of the MA segment is equal to that of the OP segment, which equals the total number of operations in the workshop, i.e., $len = \sum_{i=1}^{n} \sum_{j=1}^{I_i} J_{ij}$.

An example of two products is provided to describe this encoding scheme. The processing times and the tree-like structure of the two products are respectively presented in Table 1 and Figure 1. In Table 1, $P_i$ represents the $i$th product, $JB_{ij}$ defines the $j$th job of the $i$th product, and $O_{ijq}$ is the $q$th operation of the $j$th job in the $i$th product. In addition, '-' means that the machine is unavailable for the operation. In Figure 1, the numerical representation in each rectangle, $i$-$j$-$q$, denotes the $q$th operation of job $j$ in product $i$. Based on the above information, Figure 2 illustrates the encoding scheme of a scheduling solution. The MA segment stores the machine index following a fixed sequence of operations. Each machine is selected from the eligible machine set of the corresponding operation. The OP segment contains the codes of products and jobs. The elements with the same values belong to the same job of the product. The $r$th occurrence of the value refers to the $r$th operation of the job in the product. For example, the second '(1,3)' represents the second operation of job 3 in product 1, and the first '(2,5)' denotes the first operation of job 5 in product 2.

**Table 1.** Processing times of the operations in the two products.

| Product | Job | Operation | $M_1$ | $M_2$ | $M_3$ | $M_4$ |
|---|---|---|---|---|---|---|
| $P_1$ | $JB_{11}$ | $O_{111}$ | - | 1 | 3 | - |
|  |  | $O_{112}$ | 2 | 3 | - | 4 |
|  | $JB_{12}$ | $O_{121}$ | 2 | - | - | 3 |
|  | $JB_{13}$ | $O_{131}$ | - | 3 | 1 | 5 |
|  |  | $O_{132}$ | - | - | 5 | 2 |
|  | $JB_{14}$ | $O_{141}$ | 3 | 5 | 6 | 7 |
|  | $JB_{15}$ | $O_{151}$ | 4 | - | 3 | - |
| $P_2$ | $JB_{21}$ | $O_{211}$ | - | 1 | 2 | 5 |
|  | $JB_{22}$ | $O_{221}$ | - | 5 | 4 | 1 |
|  |  | $O_{222}$ | 4 | - | 5 | 3 |
|  | $JB_{23}$ | $O_{231}$ | 3 | 1 | 2 | 4 |
|  | $JB_{24}$ | $O_{241}$ | 5 | 4 | 7 | 3 |
|  | $JB_{25}$ | $O_{251}$ | 2 | - | 6 | 5 |
|  |  | $O_{252}$ | 3 | - | 5 | 4 |

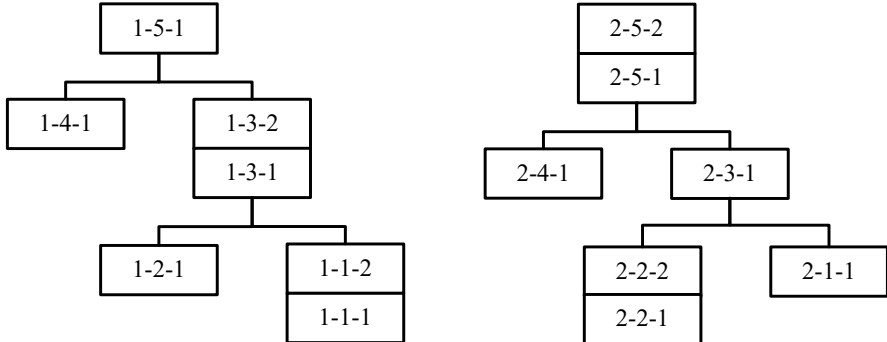

**Figure 1.** The tree-like structure of the products.

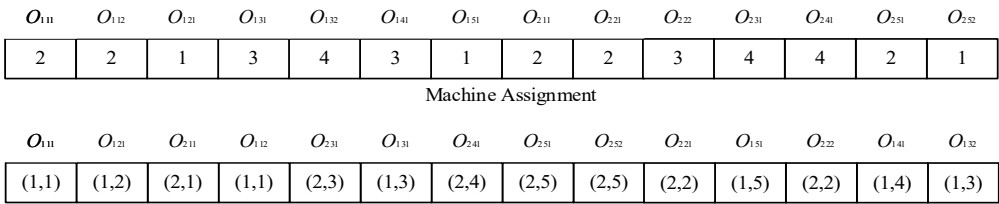

**Figure 2.** The encoding scheme.

According to the encoding scheme, the elements in the first segment are always selected from the eligible machine set, which can ensure the feasibility of the machine assignment. However, the second segment is most likely unfeasible due to the existence of hierarchical job precedence constraints. For example, $O_{132}$ and $O_{141}$ is processed before $O_{151}$ in Figure 1, which does not meet the job precedence constraints of $P_1$ in Figure 1. Here, a job precedence repair mechanism is devised to obtain a feasible operation permutation. The repair mechanism can be described below.

Step 1. Scan the OP segment of the candidate individual from left to right.

Step 2. For each operation, judge whether the current operation has precedent operations in product tree. If yes, pick out the current operation and all its precedent operations from the OP segment, keeping the positions of others unchanged.

Step 3. Relocate the current operation behind its precedent operations, and then place these elements back the OP segment following the new sequence.

After performing the repair mechanism, a feasible operation permutation is shown in Figure 3.

| $O_{111}$ | $O_{121}$ | $O_{211}$ | $O_{112}$ | $O_{241}$ | $O_{131}$ | $O_{221}$ | $O_{22}$ | $O_{231}$ | $O_{251}$ | $O_{141}$ | $O_{252}$ | $O_{132}$ | $O_{151}$ |
|---|---|---|---|---|---|---|---|---|---|---|---|---|---|
| (1,1) | (1,2) | (2,1) | (1,1) | (2,4) | (1,3) | (2,2) | (2,2) | (2,3) | (2,5) | (1,4) | (2,5) | (1,3) | (1,5) |

**Figure 3.** The OP segment after the repair operation.

### 4.2. Energy-Saving Decoding Approach

Due to job/operation precedence constraints and transportation times between machines, there exists some idle time slots between the adjacent operations on each machine. Hence, we propose an energy-saving decoding approach under the principle of the left-shift rule, which considers the makespan and energy consumption simultaneously. The left-shift rule states that if an operation can be inserted into one of the left idle time slots on the assigned machine as compactly as possible without violating the precedence constraints, the slot is first preferred. Based on this rule, the idle times of each machine are compacted, which leads to the reduction in the idle energy consumption. Meanwhile, the makespan may also be shortened, resulting in the reduction in auxiliary energy consumption. When executing the decoding approach, the OP segment of the candidate individual is traversed from left to right, and the left-shift rule is repeated for each operation until all operations in the current individual have been scheduled. For each operation, the detailed steps of the left-shift rule are shown below.

Step 1. Find out the assigned machine $k$ in the MA segment, and obtain the processing time $p_{ijqk}$ of the current operation $O_{ijq}$.

Step 2. Find out all the idle time slots in machine $k$. For each slot, it can be represented by $[t_k^S, t_k^E]$, where $t_k^S$ and $t_k^E$ are the start time and the end time of the slot, respectively.

Step 3. Traverse all the slots on machine $k$ from left to right, and try to insert the current operation $O_{ijq}$ into one slot as early as possible. When $ST_{ijq} + p_{ijqk} \leq t_k^E$ is met, the time slot is available for $O_{ijq}$. If $O_{ijq}$ has precedent operations, $ST_{ijq} = \max\left\{ t_k^S, \max\left\{ CT_{i'j'q'} + TT_{i'j'q'w,ijk} \middle| O_{i'j'q'} \in Sub_{ijq} \right\} \right\}$, $Sub_{ijq}$ is the precedent operation set of $O_{ijq}$, otherwise, $ST_{ij} = t_k^S$.

Step 4. If all slots on machine $k$ are unavailable for $O_{ijq}$, append it at the rear of machine $k$.

Figure 4 illustrates the left-shift rule, where four operations have been scheduled, i.e., $O_{111}, O_{112}, O_{121}, O_{141}$. It is assumed that $O_{131}$ is assigned to Machine 2 for processing. If the idle time slots are not considered, $O_{131}$ will be placed behind $O_{141}$. When the condition $\max\{t_2^S, \max\{CT_{112} + TT_{1123,1122}, CT_{121} + TT_{1211,1212}\}\} + p_{1312} < t_2^E$ is met, $O_{131}$ can be shifted to the left of $O_{141}$, by which the idle time slot can be compacted. This mechanism provides an opportunity for decreasing the makespan and the total energy consumption.

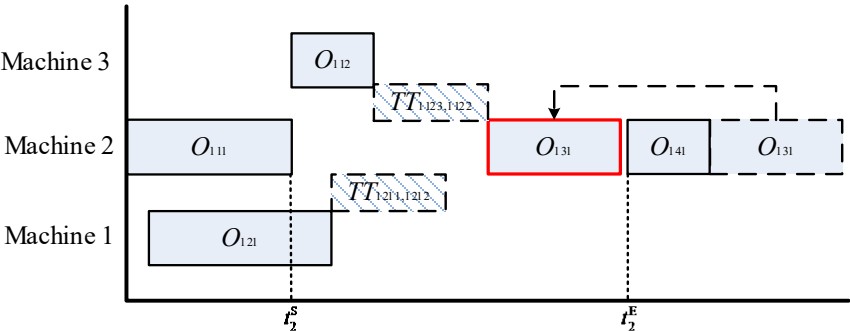

**Figure 4.** The left-shift process of operation $O_{131}$.

### 4.3. Population Initialization and Splitting

In order to obtain initial solutions with a certain quality and diversity, a population initialization approach is proposed to construct the MA and OP segments, which is described as below.

For the MA segment, the two assignment rules proposed by Pezzella et al. [32], named *AssignmentRule1* and *AssignmentRule2*, are used to select an appropriate machine for each operation. In these two rules, a matrix with the size of $len \times m$ is created to record the workload of each machine. For each operation, the machine assignment procedure includes finding the machine with the minimum workload, fixing that assignment, and then updating the workload of the selected machine by adding the processing time of the operation to all the entries in the columns where the selected machine resides. In this paper, by considering the optimization objective, the matrix is modified to record the processing energy of each machine. In addition, the random assignment rule (RAR) is also used to select a machine from the eligible machine set of each operation at random. Here, 10% of the initial population is generated by RAR, 10% by *AssignmentRule1*, and 80% by *AssignmentRule2*.

For the OP segment, we present three dispatching rules to generate the operation sequence on each machine, i.e., LLC + MWR, LLC + MOR, RSR. RSR denotes the random sequencing rule, by which the operations are randomly sequenced on each machine. The operation's lowest level (LLC) means that the operation located on the lowest level of product three has the highest priority to be scheduled. For example, in Figure 1, $O_{111}, O_{221}$ are located on the lowest level of the product tree, which have the highest priority among all the operations, $O_{121}, O_{112}$ have higher priority than $O_{131}$, etc. Most Work Remaining (MWR) defines that the operation belonging to the job with the most remaining processing time has higher priority to be selected. Most Operations Remaining (MOR) represents that the operation belonging to the job with the most remaining operations has a higher priority to be selected. LLC + MWR means that the LLC rule is first used to selected the operations with the lowest level, and the MWR rule is adopted to break a tie if more than one operation has the lowest level. LLC + MOR means that the LLC rule is first used to selected the operations with the lowest level, and the MOR rule is employed to break a tie. If one operation has been scheduled, the operation will be deleted from the product tree to update the current lowest level. Here, 40% operation sequences are generated by the LLC + MWR rule, 40% by the LLC + MOR rule, and 20% by the RSR rule.

In EHO, before undertaking the evolutionary process, the population is first split into a fixed number of clans, which can also be viewed as sub-populations. The existing research reported that the multi-population method is one of the most effective methods to maintain population diversity, by which solutions are scattered over the search space rather than focusing on a specific area [33,34]. The procedure of the population splitting is as follows: First, all solutions are sorted in descending order based on their fitness values. Then, the first elephant is assigned to the first clan, the second elephant is assigned to the second clan, the $nclan$th elephant goes to the $nclan$th clan, the $(nclan + 1)$th elephant goes to the first clan, the $(nclan + 2)$th elephant goes to the second clan, etc.

### 4.4. Clan Updating Operator

It is obvious that Equations (17) and (18) are not suitable for the discrete scheduling problem. Therefore, in this paper, the original clan updating operator is discretized based on the crossover operation. Before performing the crossover operation, a random number $rand$ is generated with a uniform distribution in [0, 1]. If $rand$ is smaller than the crossover rate $p_c$, the crossover operator will be performed to obtain a new individual.

For the candidate individual $i$ in clan $c$, the crossover operator is first performed between $X_{ci}$ and $X_{c,best}$. If the new individual $X'_{ci}$ is better than $X_{ci}$, $X_{ci}$ is replaced by $X'_{ci}$. Otherwise, the crossover operator is then performed between $X_{ci}$ and the global best individual $X_{g,best}$ in the whole population to obtain $X''_{ci}$. If $X''_{ci}$ is superior to $X_{ci}$, $X_{ci}$ is replaced by $X''_{ci}$, else $X_{ci}$ is unchanged.

For the matriarch in each clan, the crossover operator is first performed between $X_{c,best}$ and $X_{c,center}$ to generate a new individual $X'_{c,best}$. Here, $X_{c,center}$ is the elephant individual whose fitness value is closest to the mean fitness of the clan $c$. If the new individual $X'_{c,best}$ is better than $X_{c,best}$, $X_{c,best} \leftarrow X'_{c,best}$. Otherwise, the crossover operator is then performed between $X_{c,best}$ and $X_{g,best}$ to obtain $X''_{c,best}$. If $X''_{c,best}$ is superior to $X_{c,best}$, $X_{c,best}$ is replaced by $X''_{c,best}$, else $X_{c,best}$ is unchanged.

In this work, two different crossover operations are adopted for operation permutation and machine assignment, respectively. For the machine assignment, the multi-point crossover (MPX) [25] can be directly adopted. For the operation permutation, an operation is first randomly selected, which has precedent operations in product tree. For example, in Figure 5, $O_{251}$ is selected from the operation permutation. Then the precedent operations of $O_{251}$ are $O_{231}, O_{241}, O_{221}, O_{222}, O_{211}$. When performing the crossover operation, all the precedent operations $O_{231}, O_{241}, O_{221}, O_{222}, O_{211}$ of the selected operation in the two parent individuals are swapped to obtain two child individuals. The better one is selected as the new individual of the clan updating operator.

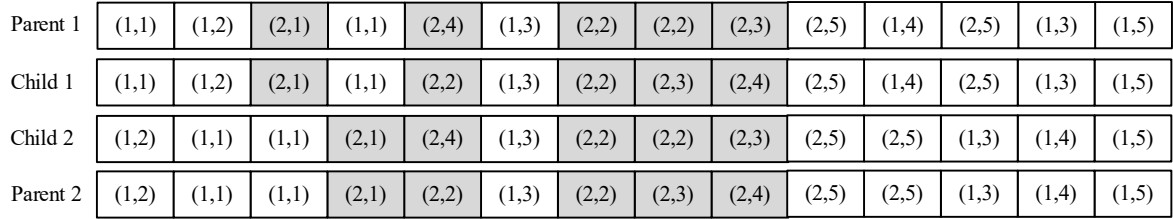

**Figure 5.** The crossover operation of the operation permutation.

### 4.5. Separating Operator

In the basic EHO algorithm, the worst elephant in each clan will be randomly replaced, as expressed in Equation (20). However, the original separation operator ignores the evaluation of the newborn calf, by which an inferior solution might join the clan. Therefore, a new separation method proposed by Li et al. [35] is employed in this section. In this method, if it is better than the original one, it can be accepted. Otherwise, a probability value $p_b$ is employed to determine whether the original elephant should be replaced. To

this end, a random number *rand* is randomly generated. If *rand* is greater than $p_b$, the original elephant will be replaced.

To implement the separating strategy, two types of neighborhood structures are proposed for the operation permutation and the machine assignment, respectively. When performing the separating operator, a neighborhood structure is randomly selected from each of Type 1 and Type 2 to acquire a new individual.

Type 1: Neighborhood structures for machine assignment

NMS1: Randomly select an operation, and then a different machine is randomly selected from the eligible machine set of the selected operation to replace the original one.

NMS2: Randomly select an operation, and then the machine with the shortest processing time is selected from the eligible machine set of the selected operation to replace the original one.

NMS3: Randomly select an operation, and then the machine with the smallest processing energy consumption coefficient is selected from the eligible machine set of the selected operation to replace the original one.

Type 2: Neighborhood structures for operation permutation

NOP1: Randomly select two elements with different values in the OP segment, and then swap the positions of the selected elements. Perform the repair mechanism to ensure the feasibility of the scheduling solution.

NOP2: Randomly select two elements with different values in the OP segment, and then insert the second element to the front of the first one. Perform the repair mechanism to ensure the feasibility of the scheduling solution.

NOP3: Randomly select two elements with different values in the OP segment, and then invert the original order of the elements between the selected two elements. Perform the repair mechanism to ensure the feasibility of the scheduling solution.

### 4.6. Steps of the IEHO

The steps of the IEHO algorithm can be summarized as below.

Step 1. Generate the initial population following the method in Section 3.4, and set some related parameters, such as the population size *popsize*, the number of clans *nclan*, the number of the saved elephants *nKEL*, the maximum generation *maxiter*, the crossover rate $p_c$ and the acceptance probability $p_b$.

Step 2. Evaluate the fitness of each elephant individual.

Step 3. Sort all the individuals in the population according to the fitness, save the *nKEL* best elephants, and then split the population into *nclan* clans with the same size.

Step 4. Perform the clan updating operation in Section 4.4.

Step 5. Perform the separating operation based on the neighborhood structure in Section 4.5.

Step 6. Merge the individuals of each clan into the population, and evaluate the fitness of all individuals.

Step 7. Perform the elitism strategy to replace the worst individuals with the *nKEL* saved ones.

Step 8. Judge whether the maximum iteration number is met. If yes, go to Step 9, otherwise, go to Step 3.

Step 9. Terminate the algorithm.

### 4.7. Time Complexity

The time complexity of the IEHO algorithm is analyzed following the above steps. Obviously, evaluate the fitness of each elephant individual in Step 2 with time complexity $O(popsize)$. Sort all the individuals in the population in Step 3 with time complexity $O(popsize)$. In Step 4, execute the clan-updating operator for all clans with time complexity $O(popsize \times 2len)$. In Step 5, perform the separating operator for all clans with time complexity $O(popsize)$. In Step 6, evaluate the fitness of all individuals with time complexity $O(popsize)$. Perform the elitism strategy to replace the worst individuals in Step 7 with

time complexity $O(nKEL)$. After omitting the low-order terms, the total time complexity of the IEHO algorithm is $O(maxiter \times popsize \times l)$, which is only related to *maxiter*, *popsize*, and *len*. *len* is the total number of operations in the workshop.

## 5. Numerical Experiments

This section aims to verify the performance of the proposed IEHO by conducting extensive experiments. In the following subsections, experimental instances and parameter settings are first elaborately designed. Then, the effectiveness of improvement strategies including population initialization strategy and energy-saving decoding approach are validated. Subsequently, IEHO is compared with some published algorithms. Finally, statistical analysis experiments are conducted to ensure the comparison results more convincingly. All algorithms were coded in Fortran language and compiled using Compaq Visual Fortran 6.6 on VMware Workstation with 6 GB RAM under Windows XP. Each experiment was conducted 10 independent times on each experimental instance for each algorithm.

### 5.1. Test Instance

Four types of different products need to be processed in a workshop, whose tree-like structures are shown in Figure 6. Thirty-two different instances are designed based on different product mixtures and the number of machines in Table 2. For each instance, some related parameters are randomly generated in a given range following a uniform distribution as follows: $nop \in [2, m]$, $p_{ijqk} \in [10, 20]$, $PE_{ijqk} \in [10, 15]$, $IE_k \in [6, 10]$, $AE \in [12, 18]$, $TE \in [5, 10]$ and $TT_{ijqw,P(ijq)k} \in [5, 15]$. *nop* represents the size of the eligible machine set for each operation.

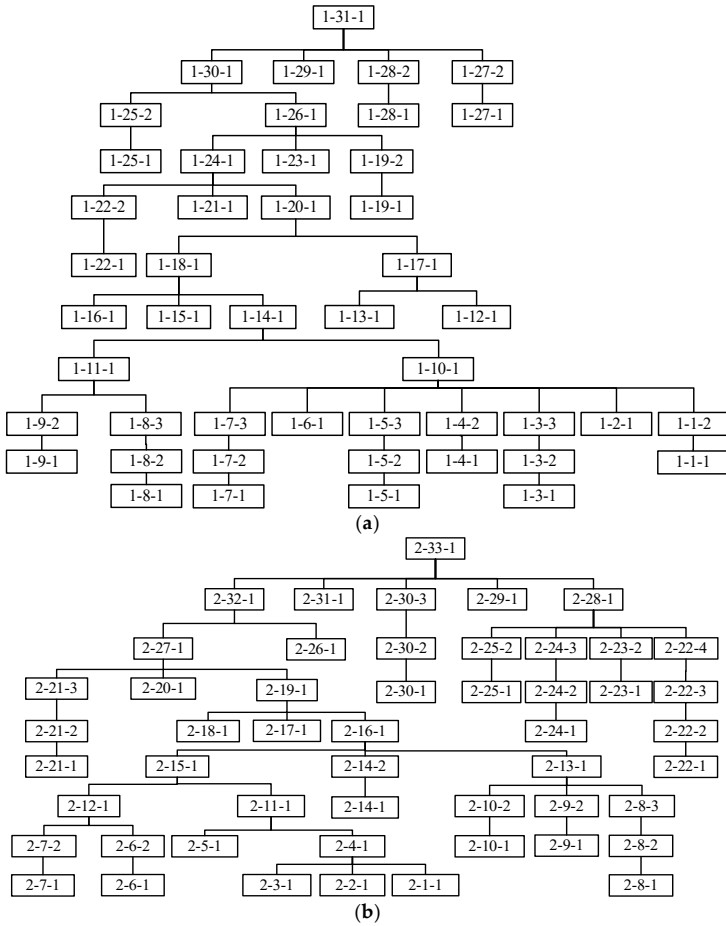

**Figure 6.** *Cont.*

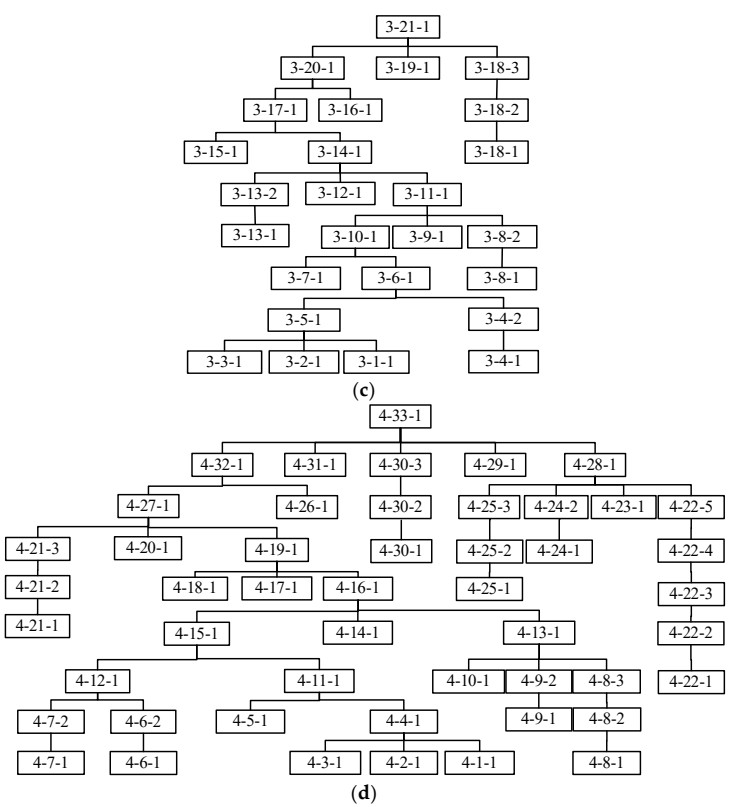

**Figure 6.** The tree-like structure of the four products. (**a**) Product 1; (**b**) Product 2; (**c**) Product 3; (**d**) Product 4.

**Table 2.** Experimental instance.

| Instance | Product Mixture | | | | *m* | Instance | Product Mixture | | | | *m* |
|---|---|---|---|---|---|---|---|---|---|---|---|
| | **P1** | **P2** | **P3** | **P4** | | | **P1** | **P2** | **P3** | **P4** | |
| RM01 | | | | | 10 | RM17 | | | | | 10 |
| RM02 | | | | | 15 | RM18 | | | | | 15 |
| RM03 | 4 | 0 | 0 | 0 | 20 | RM19 | 1 | 1 | 1 | 1 | 20 |
| RM04 | | | | | 25 | RM20 | | | | | 25 |
| RM05 | | | | | 10 | RM21 | | | | | 10 |
| RM06 | | | | | 15 | RM22 | | | | | 15 |
| RM07 | 0 | 4 | 0 | 0 | 20 | RM23 | 2 | 0 | 1 | 1 | 20 |
| RM08 | | | | | 25 | RM24 | | | | | 25 |
| RM09 | | | | | 10 | RM25 | | | | | 10 |
| RM10 | | | | | 15 | RM26 | | | | | 15 |
| RM11 | 0 | 0 | 4 | 0 | 20 | RM27 | 3 | 0 | 1 | 0 | 20 |
| RM12 | | | | | 25 | RM28 | | | | | 25 |
| RM13 | | | | | 10 | RM29 | | | | | 10 |
| RM14 | | | | | 15 | RM30 | | | | | 15 |
| RM15 | 0 | 0 | 0 | 4 | 20 | RM31 | 1 | 1 | 2 | 0 | 20 |
| RM16 | | | | | 25 | RM32 | | | | | 25 |

### 5.2. Parameter Setting

It is well-known that parameter settings impact the performance of intelligence algorithms. To set the parameters, the famous Taguchi designed experiment (DOE) method is adopted in this section. In the proposed IEHO, there are six key parameters to be set as factors, i.e., the size of the whole elephant population *popsize*, the number of clans *nclan*, the number of the saved elephants *nKEL*, the maximum generation *maxiter*, the crossover rate $p_c$, and the acceptance probability $p_b$. For each factor, five levels are considered as

reported in Table 3. Then, the orthogonal array $L_{25}(5^6)$ is constructed based on the instance RM17 in Table 4, where *Avg* represents the mean value of *TEC* in ten independent runs. The response value and the significance rank are reported in Table 5. According to the simulation data, the six parameters are set following the trend of factor level in Figure 7, i.e., $PS = 300$, $nclan = 4$, $maxiter = 500$, $nKEL = 6$, $p_c = 0.9$, $p_b = 0.7$.

**Table 3.** Parameter levels.

| Factor 1~6 | Level 1~5 | | | | |
|---|---|---|---|---|---|
| *PS* | 60 | 120 | 180 | 240 | 300 |
| *nclan* | 1 | 2 | 3 | 4 | 5 |
| *maxiter* | 100 | 200 | 300 | 400 | 500 |
| *nKEL* | 2 | 4 | 6 | 8 | 10 |
| $p_c$ | 0.5 | 0.6 | 0.7 | 0.8 | 0.9 |
| $p_b$ | 0.5 | 0.6 | 0.7 | 0.8 | 0.9 |

**Table 4.** Orthogonal array and Avg values.

| Combination Number | Factor 1~6 | | | | | | Avg |
|---|---|---|---|---|---|---|---|
| | *PS* | *nclan* | *maxiter* | *nKEL* | $p_c$ | $p_b$ | |
| 1 | 1 | 1 | 1 | 1 | 1 | 1 | 40,038.4 |
| 2 | 1 | 2 | 2 | 2 | 2 | 2 | 39,729.9 |
| 3 | 1 | 3 | 3 | 3 | 3 | 3 | 39,374.9 |
| 4 | 1 | 4 | 4 | 4 | 4 | 4 | 39,306.3 |
| 5 | 1 | 5 | 5 | 5 | 5 | 5 | 39,205.4 |
| 6 | 2 | 1 | 2 | 3 | 4 | 5 | 39,630.0 |
| 7 | 2 | 2 | 3 | 4 | 5 | 1 | 39,408.6 |
| 8 | 2 | 3 | 4 | 5 | 1 | 2 | 39,446.1 |
| 9 | 2 | 4 | 5 | 1 | 2 | 3 | 38,991.9 |
| 10 | 2 | 5 | 1 | 2 | 3 | 4 | 39,404.1 |
| 11 | 3 | 1 | 3 | 5 | 2 | 4 | 39,477.2 |
| 12 | 3 | 2 | 4 | 1 | 3 | 5 | 39,276.1 |
| 13 | 3 | 3 | 5 | 2 | 4 | 1 | 39,516.0 |
| 14 | 3 | 4 | 1 | 3 | 5 | 2 | 39,258.7 |
| 15 | 3 | 5 | 2 | 4 | 1 | 3 | 39,191.6 |
| 16 | 4 | 1 | 4 | 2 | 5 | 3 | 39,405.4 |
| 17 | 4 | 2 | 5 | 3 | 1 | 4 | 39,224.4 |
| 18 | 4 | 3 | 1 | 4 | 2 | 5 | 39,356.0 |
| 19 | 4 | 4 | 2 | 5 | 3 | 1 | 39,088.5 |
| 20 | 4 | 5 | 3 | 1 | 4 | 2 | 39,111.3 |
| 21 | 5 | 1 | 5 | 4 | 3 | 2 | 39,460.8 |
| 22 | 5 | 2 | 1 | 5 | 4 | 3 | 39,336.3 |
| 23 | 5 | 3 | 2 | 1 | 5 | 4 | 39,155.5 |
| 24 | 5 | 4 | 3 | 2 | 1 | 5 | 39,057.1 |
| 25 | 5 | 5 | 4 | 3 | 2 | 1 | 39,027.3 |

**Table 5.** Response value and significance rank.

| Level | Factor 1~6 | | | | | |
|---|---|---|---|---|---|---|
| | *PS* | *nclan* | *maxiter* | *nKEL* | $p_c$ | $p_b$ |
| 1 | 39,531.0 | 39,602.4 | 39,478.7 | 39,314.6 | 39,391.5 | 39,415.8 |
| 2 | 39,376.1 | 39,395.1 | 39,359.1 | 39,422.5 | 39,316.5 | 39,401.4 |
| 3 | 39,343.9 | 39,369.7 | 39,285.8 | 39,303.1 | 39,320.9 | 39,260.0 |
| 4 | 39,237.1 | 39,140.5 | 39,292.2 | 39,344.7 | 39,380.0 | 39,313.5 |
| 5 | 39,207.4 | 39,187.9 | 39,279.7 | 39,310.7 | 39,286.7 | 39,304.9 |
| Delta | 323.6 | 461.9 | 199.0 | 119.4 | 104.8 | 155.8 |
| Rank | 1 | 2 | 3 | 5 | 6 | 4 |

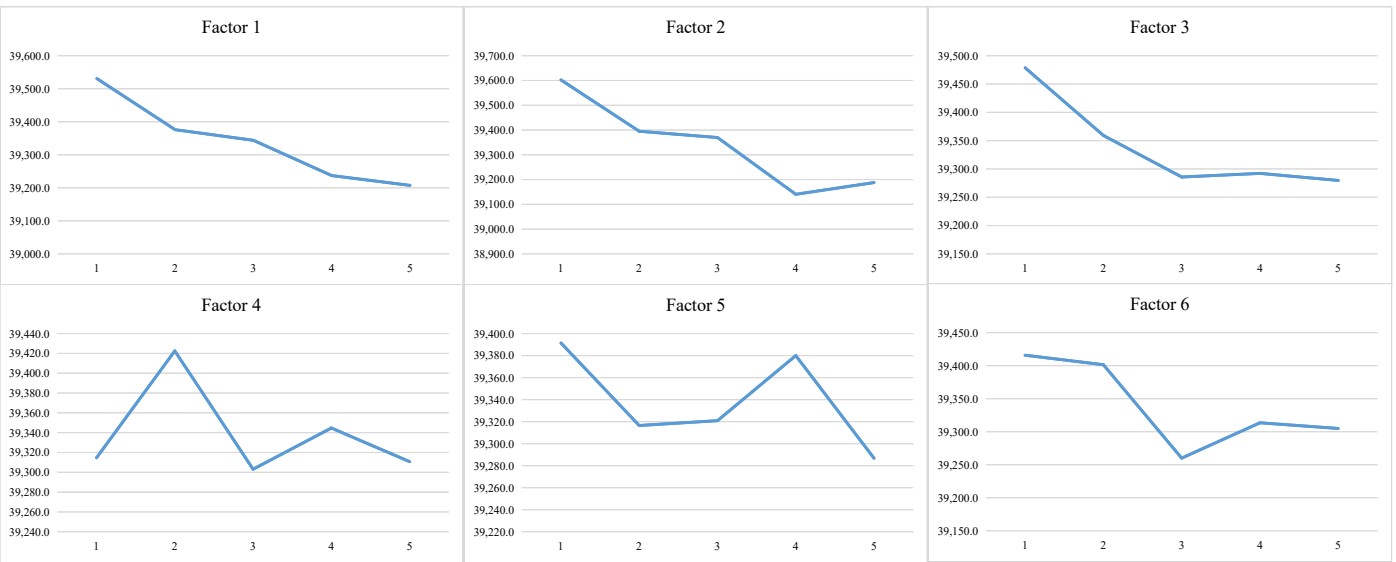

**Figure 7.** Factor level trend of parameters.

### 5.3. Comparison Results of Different Algorithms

To verify the effectiveness of the proposed algorithm, seven algorithms are compared with the IEHO algorithm, i.e., IEHONL, IEHORR, GA [24], PSO-GA [26], VNS [36], DABO [37], and CosSDCSO [8]. IEHONL and IEHORR are the abbreviated versions of the proposed IEHO. In IEHONL, the left-shift rule is not applied in the decoding approach. In IEHORR, the initial population is generated at random. The GA and the VNS were designed for the AJSP without the consideration of job transportation times and energy-related metrics. The PSO-GA was proposed for a two-stage energy-saving AJSP without the consideration of job transportation times. The DABO and the CosSDCSO were proposed for the energy-saving FJSP without considering the transportation times and the hierarchical job precedence constraints. All the compared algorithms are easily modified to adjust to the considered problem.

#### 5.3.1. Effectiveness of the Proposed Population Initialization Approach

To validate the effectiveness of the proposed population initialization approach, IEHORR is first compared with the IEHO algorithm in this section. For IEHORR, the parameters are the same as those of IEHO. The comparison results are reported in Table 6, where the first column shows the instance names, and the following columns show the experimental data of the compared algorithms. In Table 6, '*Best*' is the best objective value collected by each algorithm in ten runs. '*Avg*' denotes the average results in ten runs. '*Std*' is the standard deviation of objective value. '*Time*' is the average running time (in seconds) of each algorithm. '*Mean*' represents the average value of the corresponding indicator in the same column. It can be easily observed from Table 6 that: (1) In comparisons of *Best*, *Avg* and *Std*, IEHO outperforms IEHORR in all the instances, which indicates that IEHO has a stronger search ability and performs more stably than IEHORR. (2) In the comparison of *Time* value, there is little difference between the two algorithms. (3) The *Mean* value in the last line also demonstrates the superior performance of the IEHO algorithm. In summary, the proposed population initialization approach is effective for the considered problem.

**Table 6.** Comparison results of IEHO and IEHRR.

| Instance | IEHO | | | | IEHORR | | | |
|---|---|---|---|---|---|---|---|---|
| | *Best* | *Avg* | *Std* | *Time* | *Best* | *Avg* | *Std* | *Time* |
| RM01 | 38,929 | 39,172.4 | 115.9 | 1862.5 | 44,053 | 44,812.8 | 499.9 | 1870.6 |
| RM02 | 40,938 | 41,165.4 | 152.9 | 1492.7 | 45,537 | 46,484.6 | 518.5 | 1508.7 |
| RM03 | 39,021 | 39,457.5 | 312.4 | 1311.2 | 43,776 | 44,800.8 | 619.2 | 1328.1 |
| RM04 | 37,085 | 37,572.7 | 319.0 | 1222.2 | 41,930 | 42,945.8 | 845.8 | 1225.6 |
| RM05 | 48,178 | 48,414.5 | 171.8 | 2270.8 | 52,044 | 52,890.9 | 604.3 | 2233.2 |
| RM06 | 45,182 | 45,721.7 | 326.0 | 1712.9 | 49,957 | 50,879.6 | 593.0 | 1736.3 |
| RM07 | 42,720 | 42,993.5 | 238.3 | 1510.9 | 47,451 | 48,358.9 | 689.0 | 1497.9 |
| RM08 | 43,474 | 44,206.5 | 334.1 | 1390.9 | 47,763 | 49,031.5 | 861.0 | 1411.0 |
| RM09 | 25,671 | 25,998.4 | 197.4 | 687.2 | 27,232 | 27,752.2 | 335.9 | 694.0 |
| RM10 | 23,450 | 24,069.2 | 439.0 | 556.2 | 24,170 | 25,294.4 | 727.1 | 584.9 |
| RM11 | 23,792 | 24,228.6 | 252.5 | 521.7 | 24,800 | 25,606.3 | 448.9 | 519.7 |
| RM12 | 19,690 | 20,282.5 | 334.6 | 515.5 | 20,828 | 21,635.4 | 484.0 | 490.3 |
| RM13 | 46,292 | 46,694.4 | 243.8 | 2131.4 | 49,686 | 50,442.5 | 542.7 | 2064.2 |
| RM14 | 44,993 | 45,225.7 | 285.9 | 1692.3 | 48,954 | 49,990.5 | 691.5 | 1642.3 |
| RM15 | 37,352 | 37,949.8 | 301.9 | 1498.4 | 42,381 | 43,409.8 | 538.5 | 1466.8 |
| RM16 | 43,152 | 43,633.7 | 277.8 | 1383.4 | 47,816 | 49,153.2 | 717.7 | 1369.6 |
| RM17 | 38,890 | 39,135.1 | 172.0 | 1684.8 | 41,828 | 43,376.0 | 874.9 | 1722.8 |
| RM18 | 40,335 | 41,227.0 | 377.0 | 1376.8 | 44,402 | 45,568.8 | 769.9 | 1393.2 |
| RM19 | 37,171 | 38,024.0 | 432.4 | 1132.0 | 42,053 | 43,114.5 | 1013.1 | 1182.3 |
| RM20 | 35,477 | 36,044.8 | 405.7 | 1172.8 | 38,543 | 40,040.8 | 1366.9 | 1155.2 |
| RM21 | 36,173 | 36,408.0 | 199.3 | 1739.3 | 39,569 | 40,192.7 | 421.5 | 1746.3 |
| RM22 | 36,315 | 36,869.1 | 263.4 | 1392.5 | 40,141 | 41,240.2 | 643.3 | 1425.7 |
| RM23 | 36,598 | 36,912.5 | 185.9 | 1233.9 | 38,992 | 40,178.5 | 628.0 | 1239.9 |
| RM24 | 33,225 | 33,736.7 | 195.9 | 1148.3 | 36,353 | 37,070.1 | 645.7 | 1147.6 |
| RM25 | 38,923 | 39,080.6 | 106.9 | 1517.1 | 42,196 | 43,121.5 | 586.5 | 1572.7 |
| RM26 | 36,294 | 36,713.6 | 193.4 | 1239.6 | 39,881 | 40,688.6 | 648.7 | 1299.1 |
| RM27 | 33,863 | 34,251.4 | 210.3 | 1086.5 | 37,304 | 38,392.9 | 722.6 | 1106.4 |
| RM28 | 31,620 | 32,246.0 | 307.4 | 1013.7 | 34,585 | 35,832.5 | 829.3 | 1003.6 |
| RM29 | 32,118 | 32,467.2 | 181.0 | 1349.9 | 34,572 | 35,448.2 | 502.4 | 1334.2 |
| RM30 | 32,523 | 32,832.1 | 208.0 | 1099.0 | 35,808 | 36,608.1 | 523.3 | 1095.3 |
| RM31 | 32,485 | 32,863.8 | 278.3 | 918.6 | 35,136 | 35,750.2 | 350.1 | 897.8 |
| RM32 | 31,072 | 31,746.7 | 385.7 | 849.9 | 33,430 | 34,500.9 | 547.2 | 851.1 |
| Mean | 36,343.8 | 36,792.0 | 262.7 | 1305.1 | 39,786.6 | 40,769.2 | 649.7 | 1306.8 |

### 5.3.2. Effectiveness of the Energy-Saving Decoding Approach

To evaluate the performance of the proposed energy-saving decoding approach, IEHONL is compared with the proposed IEHO algorithm. For IEHONL, the parameters are also the same as those of IEHO. From the results in Table 7, it can be easily observed that: (1) IEHO performs better than IEHONL in comparisons of the values of *Best*, *Avg*, and *Std*, which means that IEHO has a stronger search ability and performs more stably than IEHONL. (2) For the *Time* value, IEHO takes more time than IEHONL due to the introduction of the left-shift rule in the energy-saving decoding approach. (3) The last line in Table 7 also demonstrates the superior performance of the IEHO algorithm except for the computational time. In summary, the proposed energy-saving decoding approach increases the computational time, but it can significantly improve the solution quality.

**Table 7.** Comparison results of IEHO and IEHNL.

| Instance | IEHO | | | | IEHONL | | | |
|---|---|---|---|---|---|---|---|---|
| | *Best* | *Avg* | *Std* | *Time* | *Best* | *Avg* | *Std* | *Time* |
| RM01 | 38,929 | 39,172.4 | 115.9 | 1862.5 | 41,085 | 41,597.4 | 370.4 | 255.1 |
| RM02 | 40,938 | 41,165.4 | 152.9 | 1492.7 | 43,521 | 44,097.2 | 351.8 | 262.3 |
| RM03 | 39,021 | 39,457.5 | 312.4 | 1311.2 | 42,512 | 42,909.3 | 215.1 | 274.0 |
| RM04 | 37,085 | 37,572.7 | 319.0 | 1222.2 | 41,370 | 42,558.3 | 519.0 | 281.8 |
| RM05 | 48,178 | 48,414.5 | 171.8 | 2270.8 | 49,394 | 50,292.5 | 439.5 | 289.9 |
| RM06 | 45,182 | 45,721.7 | 326.0 | 1712.9 | 48,433 | 49,339.7 | 720.2 | 299.0 |
| RM07 | 42,720 | 42,993.5 | 238.3 | 1510.9 | 45,786 | 46,938.4 | 766.8 | 310.8 |
| RM08 | 43,474 | 44,206.5 | 334.1 | 1390.9 | 46,220 | 47,890.9 | 729.6 | 323.3 |
| RM09 | 25,671 | 25,998.4 | 197.4 | 687.2 | 26,882 | 27,926.2 | 656.6 | 110.8 |
| RM10 | 23,450 | 24,069.2 | 439.0 | 556.2 | 25,131 | 26,177.2 | 491.9 | 115.8 |
| RM11 | 23,792 | 24,228.6 | 252.5 | 521.7 | 25,351 | 26,453.3 | 456.5 | 120.8 |
| RM12 | 19,690 | 20,282.5 | 334.6 | 515.5 | 21,540 | 22,459.3 | 529.3 | 127.0 |
| RM13 | 46,292 | 46,694.4 | 243.8 | 2131.4 | 48,206 | 49,282.4 | 686.6 | 282.6 |
| RM14 | 44,993 | 45,225.7 | 285.9 | 1692.3 | 47,637 | 48,988.9 | 942.4 | 290.6 |
| RM15 | 37,352 | 37,949.8 | 301.9 | 1498.4 | 40,899 | 42,152.9 | 596.0 | 302.7 |
| RM16 | 43,152 | 43,633.7 | 277.8 | 1383.4 | 47,581 | 48,627.9 | 723.4 | 312.3 |
| RM17 | 38,890 | 39,135.1 | 172.0 | 1684.8 | 41,023 | 42,289.7 | 1060.1 | 254.8 |
| RM18 | 40,335 | 41,227.0 | 377.0 | 1376.8 | 44,703 | 45,733.8 | 767.2 | 259.4 |
| RM19 | 37,171 | 38,024.0 | 432.4 | 1132.0 | 43,276 | 43,727.7 | 300.8 | 274.2 |
| RM20 | 35,477 | 36,044.8 | 405.7 | 1172.8 | 40,155 | 41,292.4 | 708.3 | 282.0 |
| RM21 | 36,173 | 36,408.0 | 199.3 | 1739.3 | 38,687 | 39,253.0 | 344.7 | 241.3 |
| RM22 | 36,315 | 36,869.1 | 263.4 | 1392.5 | 39,427 | 40,627.8 | 542.9 | 249.7 |
| RM23 | 36,598 | 36,912.5 | 185.9 | 1233.9 | 39,684 | 40,779.8 | 727.8 | 260.8 |
| RM24 | 33,225 | 33,736.7 | 195.9 | 1148.3 | 36,946 | 37,891.2 | 668.9 | 271.6 |
| RM25 | 38,923 | 39,080.6 | 106.9 | 1517.1 | 40,165 | 40,883.5 | 423.4 | 210.0 |
| RM26 | 36,294 | 36,713.6 | 193.4 | 1239.6 | 38,529 | 39,460.1 | 577.8 | 217.3 |
| RM27 | 33,863 | 34,251.4 | 210.3 | 1086.5 | 36,974 | 37,974.2 | 576.7 | 228.0 |
| RM28 | 31,620 | 32,246.0 | 307.4 | 1013.7 | 34,440 | 35,573.4 | 670.9 | 236.2 |
| RM29 | 32,118 | 32,467.2 | 181.0 | 1349.9 | 34,660 | 35,461.6 | 479.4 | 177.9 |
| RM30 | 32,523 | 32,832.1 | 208.0 | 1099.0 | 36,476 | 37,397.8 | 388.3 | 184.6 |
| RM31 | 32,485 | 32,863.8 | 278.3 | 918.6 | 36,384 | 36,872.8 | 413.7 | 194.4 |
| RM32 | 31,072 | 31,746.7 | 385.7 | 849.9 | 34,634 | 35,420.4 | 543.2 | 200.6 |
| Mean | 36,343.8 | 36,792.0 | 262.7 | 1305.1 | 39,303.5 | 40,260.3 | 574.7 | 240.7 |

### 5.3.3. Comparison with Other Algorithms

To further demonstrate the advantage of the proposed IEHO algorithm in solving the considered problem, we compared IEHO with five published algorithms, namely, GA [24], PSO-GA [26], VNS [36], DABO [37], and CosSDCSO [8]. The parameters of these algorithms are as follows: In GA, the population size is 300, the crossover rate is 0.8, and the mutate rate is 0.1. In PSO-GA, the population size is 300, the crossover rate is 0.9, and the mutation rate is 0.1. In DABO, the population size is 300, the lifespan of the population is 5, the crossover rate is 0.8, and the mutation rate is 0.1. In CosSDCSO, the population size is 300, the crossover rate is 0.7, the size of memory pool is 15. To make a fair comparison, the termination conditions of the compared algorithms are set to the same running time as the IEHO algorithm in Tables 6 and 7. According to the comparison results in Table 8, the following observations can be obtained: (1) Comparison of the values of *Best* and *Avg* shows that IEHO outperforms other algorithms in all instances, which demonstrates that IEHO has a stronger search ability than others. (2) In the comparison of the value of *Std*, IEHO performs best in 22 out of 32 instances, which demonstrates that IEHO is more stable than other algorithms. The second best algorithm is CosSDCSO, which obtains the best values of *Std* in 10 out of 32 instances. (3) The last row suggests that, on average, the proposed algorithm performs better than the other algorithms. In addition, Figure 8 shows the convergence curve of each algorithm for 12 different instances. It can be easily observed that the convergence speed of IEHO is faster than that of the other compared algorithms. Meanwhile, the search capabilities of IEHO are stronger than those of the others. In summary, the proposed IEHO is significantly superior to other algorithms for solving the considered problem.

**Table 8.** Comparison results of IEHO and other algorithms.

| Instance | IEHO | | | GA | | | PSO-GA | | |
|---|---|---|---|---|---|---|---|---|---|
| | *Best* | *Avg* | *Std* | *Best* | *Avg* | *Std* | *Best* | *Avg* | *Std* |
| RM01 | 38,929 | 39,172.4 | 115.9 | 49,330 | 50,458.9 | 648.0 | 41,191 | 42,117.0 | 553.0 |
| RM02 | 40,938 | 41,165.4 | 152.9 | 52,828 | 54,209.4 | 812.0 | 43,947 | 44,893.1 | 544.1 |
| RM03 | 39,021 | 39,457.5 | 312.4 | 52,547 | 54,272.8 | 1094.3 | 42,936 | 43,953.9 | 573.9 |
| RM04 | 37,085 | 37,572.7 | 319.0 | 53,084 | 54,452.8 | 798.2 | 43,602 | 44,681.5 | 872.6 |
| RM05 | 48,178 | 48,414.5 | 171.8 | 58,173 | 59,267.7 | 737.6 | 49,256 | 50,479.8 | 694.3 |
| RM06 | 45,182 | 45,721.7 | 326.0 | 57,317 | 59,842.7 | 1118.6 | 48,038 | 49,742.3 | 885.4 |
| RM07 | 42,720 | 42,993.5 | 238.3 | 57,509 | 59,016.5 | 948.4 | 46,737 | 47,948.7 | 989.5 |
| RM08 | 43,474 | 44,206.5 | 334.1 | 59,880 | 61,491.2 | 1149.7 | 48,100 | 49,029.9 | 654.8 |
| RM09 | 25,671 | 25,998.4 | 197.4 | 30,296 | 31,301.2 | 666.5 | 27,569 | 27,877.2 | 221.1 |
| RM10 | 23,450 | 24,069.2 | 439.0 | 28,592 | 29,329.7 | 628.8 | 24,979 | 25,769.3 | 498.5 |
| RM11 | 23,792 | 24,228.6 | 252.5 | 29,893 | 30,682.4 | 572.6 | 26,150 | 27,012.4 | 347.1 |
| RM12 | 19,690 | 20,282.5 | 334.6 | 25,468 | 26,509.5 | 605.2 | 21,548 | 22,266.1 | 413.4 |
| RM13 | 46,292 | 46,694.4 | 243.8 | 56,112 | 57,357.0 | 731.7 | 48,322 | 50,423.0 | 1027.9 |
| RM14 | 44,993 | 45,225.7 | 285.9 | 57,275 | 58,955.2 | 768.1 | 47,984 | 49,781.1 | 999.3 |
| RM15 | 37,352 | 37,949.8 | 301.9 | 52,166 | 52,892.1 | 575.8 | 41,189 | 42,635.5 | 921.6 |
| RM16 | 43,152 | 43,633.7 | 277.8 | 59,797 | 60,488.4 | 671.8 | 48,845 | 49,851.1 | 697.7 |
| RM17 | 38,890 | 39,135.1 | 172.0 | 47,304 | 48,435.2 | 619.5 | 40,714 | 41,847.2 | 654.6 |
| RM18 | 40,335 | 41,227.0 | 377.0 | 50,502 | 51,942.8 | 770.7 | 44,690 | 45,254.2 | 408.4 |
| RM19 | 37,171 | 38,024.0 | 432.4 | 50,024 | 50,930.3 | 524.0 | 43,393 | 44,197.3 | 704.1 |
| RM20 | 35,477 | 36,044.8 | 405.7 | 48,632 | 49,927.9 | 944.3 | 40,511 | 41,757.8 | 699.7 |
| RM21 | 36,173 | 36,408.0 | 199.3 | 44,323 | 45,503.8 | 714.3 | 38,278 | 39,368.9 | 578.8 |
| RM22 | 36,315 | 36,869.1 | 263.4 | 46,342 | 48,305.9 | 947.0 | 40,609 | 41,513.9 | 545.5 |
| RM23 | 36,598 | 36,912.5 | 185.9 | 47,079 | 48,190.4 | 878.8 | 40,916 | 42,042.5 | 632.9 |
| RM24 | 33,225 | 33,736.7 | 195.9 | 44,398 | 45,835.1 | 832.1 | 37,848 | 39,833.9 | 1038.4 |
| RM25 | 38,923 | 39,080.6 | 106.9 | 46,743 | 47,979.5 | 543.6 | 40,586 | 41,224.7 | 437.2 |
| RM26 | 36,294 | 36,713.6 | 193.4 | 46,425 | 47,661.2 | 753.4 | 38,570 | 39,488.2 | 482.0 |
| RM27 | 33,863 | 34,251.4 | 210.3 | 44,001 | 45,568.7 | 813.4 | 37,321 | 39,175.9 | 808.0 |
| RM28 | 31,620 | 32,246.0 | 307.4 | 42,558 | 44,609.2 | 1189.7 | 35,238 | 36,911.3 | 1234.1 |
| RM29 | 32,118 | 32,467.2 | 181.0 | 39,093 | 40,306.6 | 651.4 | 34,381 | 35,055.6 | 538.1 |
| RM30 | 32,523 | 32,832.1 | 208.0 | 42,444 | 43,104.5 | 433.1 | 36,222 | 37,637.3 | 734.8 |
| RM31 | 32,485 | 32,863.8 | 278.3 | 41,772 | 43,221.7 | 771.3 | 36,118 | 37,642.0 | 1058.4 |
| RM32 | 31,072 | 31,746.7 | 385.7 | 42,089 | 43,859.5 | 727.0 | 35,249 | 37,208.3 | 905.0 |
| Mean | 36,343.8 | 36,792.0 | 262.7 | 46,999.9 | 48,309.7 | 770.0 | 39,719.9 | 40,894.4 | 698.6 |

| Instance | VNS | | | DABO | | | CosSDCSO | | |
|---|---|---|---|---|---|---|---|---|---|
| | *Best* | *Avg* | *Std* | *Best* | *Avg* | *Std* | *Best* | *Avg* | *Std* |
| RM01 | 43,113 | 44,196.2 | 531.2 | 41,615 | 42,590.7 | 764.1 | 42,212 | 42,727.4 | 336.0 |
| RM02 | 46,437 | 47,802.9 | 1027.7 | 43,190 | 44,237.2 | 626.9 | 45,621 | 45,858.7 | 175.7 |
| RM03 | 45,315 | 46,358.9 | 882.9 | 43,187 | 43,937.6 | 710.4 | 44,825 | 45,817.7 | 414.7 |
| RM04 | 45,475 | 47,845.6 | 1088.4 | 41,314 | 42,685.7 | 1458.9 | 45,909 | 46,837.3 | 413.9 |
| RM05 | 50,919 | 52,157.6 | 672.9 | 48,335 | 50,294.7 | 768.2 | 50,326 | 50,698.1 | 228.2 |
| RM06 | 50,608 | 51,792.5 | 732.4 | 47,433 | 48,142.2 | 386.6 | 49,082 | 49,695.6 | 284.0 |
| RM07 | 49,030 | 50,332.7 | 761.7 | 44,342 | 44,823.7 | 480.5 | 46,735 | 47,269.8 | 327.8 |
| RM08 | 50,388 | 51,097.9 | 525.1 | 45,316 | 46,151.8 | 475.7 | 48,939 | 49,306.9 | 258.3 |
| RM09 | 27,779 | 29,083.6 | 684.0 | 26,774 | 27,305.5 | 247.6 | 27,832 | 28,133.3 | 137.3 |
| RM10 | 26,701 | 27,730.7 | 404.7 | 24,559 | 25,137.5 | 410.4 | 27,222 | 27,600.0 | 210.4 |
| RM11 | 27,433 | 28,282.6 | 545.0 | 24,540 | 25,367.1 | 375.1 | 27,935 | 28,454.4 | 196.9 |
| RM12 | 23,822 | 24,462.0 | 356.2 | 20,811 | 21,439.4 | 334.9 | 24,678 | 25,024.6 | 199.3 |
| RM13 | 47,785 | 50,326.8 | 1013.2 | 48,322 | 49,288.8 | 804.3 | 48,485 | 48,945.3 | 282.8 |
| RM14 | 49,125 | 50,029.2 | 592.9 | 45,952 | 47,004.3 | 634.2 | 48,552 | 48,667.1 | 105.6 |
| RM15 | 44,059 | 45,047.7 | 890.2 | 39,943 | 40,391.3 | 305.5 | 42,242 | 42,998.9 | 303.5 |
| RM16 | 51,414 | 52,258.1 | 561.6 | 44,970 | 45,807.1 | 614.3 | 49,031 | 49,693.3 | 388.6 |

**Table 8.** *Cont.*

| Instance | VNS | | | DABO | | | CosSDCSO | | |
|---|---|---|---|---|---|---|---|---|---|
| | *Best* | *Avg* | *Std* | *Best* | *Avg* | *Std* | *Best* | *Avg* | *Std* |
| RM17 | 41,499 | 42,629.9 | 714.8 | 39,113 | 39,866.2 | 391.7 | 40,082 | 40,677.9 | 267.0 |
| RM18 | 44,923 | 45,723.5 | 632.8 | 41,460 | 42,150.1 | 402.0 | 43,662 | 44,248.4 | 285.1 |
| RM19 | 44,436 | 45,539.8 | 436.0 | 39,489 | 40,529.1 | 977.5 | 43,564 | 44,078.9 | 276.3 |
| RM20 | 42,234 | 43,549.8 | 931.2 | 37,696 | 38,273.9 | 436.2 | 40,316 | 40,876.5 | 296.3 |
| RM21 | 39,260 | 40,371.0 | 741.2 | 38,172 | 38,586.9 | 259.9 | 38,818 | 39,299.2 | 292.9 |
| RM22 | 41,781 | 43,212.3 | 902.8 | 38,089 | 38,927.2 | 702.1 | 39,832 | 41,070.4 | 618.1 |
| RM23 | 41,437 | 43,120.1 | 876.8 | 37,812 | 38,547.4 | 483.8 | 41,427 | 41,794.5 | 222.4 |
| RM24 | 40,095 | 40,889.6 | 484.0 | 35,237 | 36,062.9 | 446.6 | 39,120 | 39,609.9 | 295.2 |
| RM25 | 42,369 | 43,235.1 | 662.9 | 40,583 | 41,872.9 | 987.3 | 41,639 | 42,007.9 | 222.7 |
| RM26 | 40,065 | 41,932.3 | 812.9 | 38,363 | 39,250.1 | 402.3 | 40,336 | 40,766 | 272.9 |
| RM27 | 40,536 | 41,638.8 | 842.9 | 36,164 | 37,154.8 | 600.3 | 39,859 | 40,447.8 | 309.3 |
| RM28 | 36,859 | 39,121.6 | 1157.7 | 33,377 | 34,315.6 | 598.4 | 38,017 | 38,847.3 | 517.0 |
| RM29 | 35,651 | 36,568.5 | 661.8 | 33,608 | 34,155.0 | 428.8 | 34,676 | 35,019.2 | 191.0 |
| RM30 | 37,455 | 38,629.6 | 848.8 | 34,954 | 35,409.9 | 313.6 | 37,244 | 37,783.7 | 297.3 |
| RM31 | 38,069 | 39,372.2 | 1020.7 | 33,737 | 34,368.0 | 282.8 | 37,462 | 38,119.3 | 330.8 |
| RM32 | 37,484 | 38,543.6 | 904.1 | 32,681 | 33,386.2 | 574.8 | 36,199 | 36,850.2 | 418.0 |
| Mean | 41,361.1 | 42,590.1 | 746.9 | 38,160.6 | 38,983.2 | 552.7 | 40,683.7 | 41,225.8 | 293.0 |

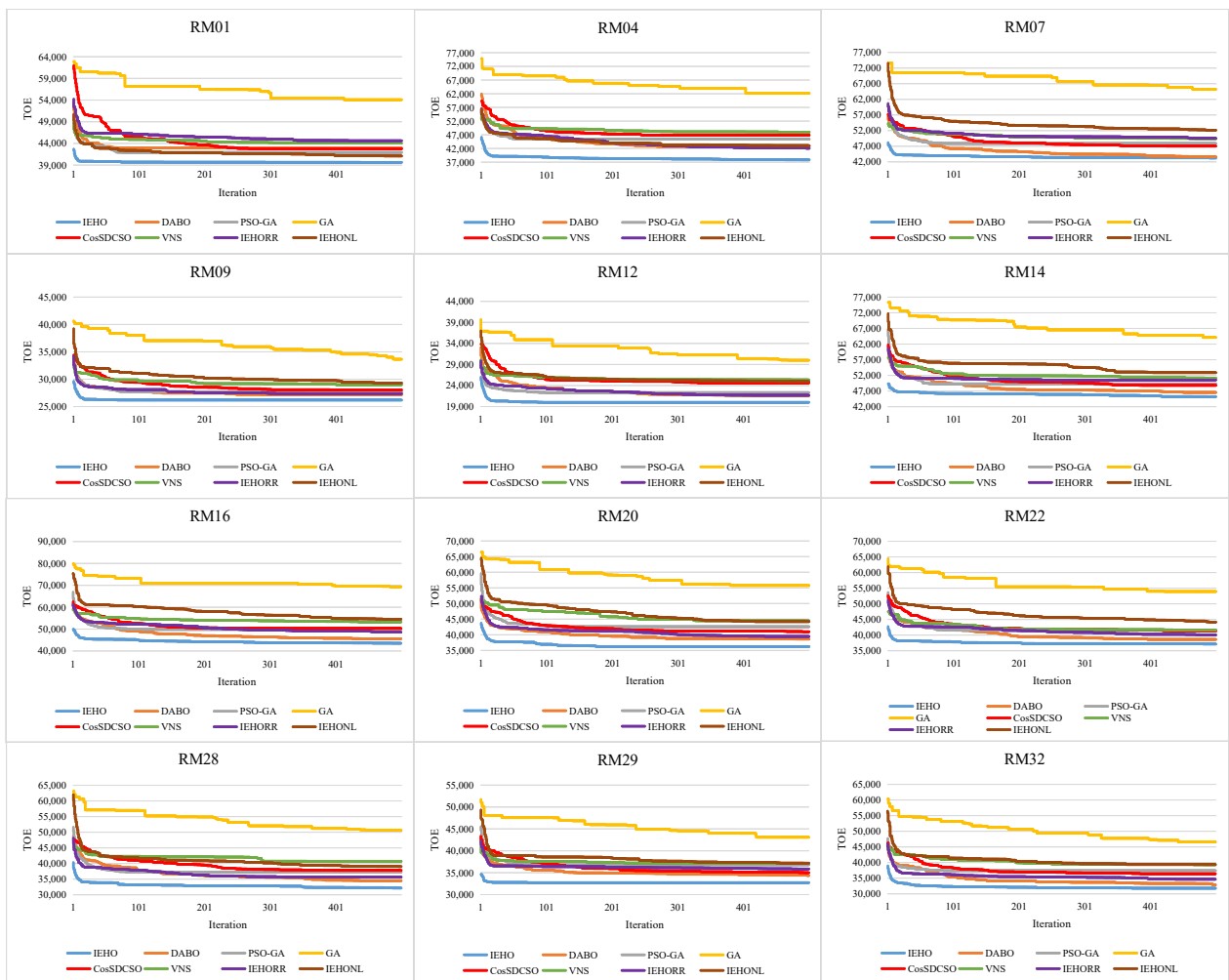

**Figure 8.** Convergence curves of different algorithms.

### 5.3.4. Statistical Analysis of the Computational Results

To ensure the comparison results are statistically more convincing, this section conducts the paired-sample *t*-test to test the performance of the algorithms. Before conducting the statistical analysis, the values of '*BRPD*' and '*ARPD*' are calculated in Table 9, i.e., $BRPD = \frac{100 \times (Best - Gbest)}{Gbest}$, $ARPD = \frac{100 \times (Avg - Gbest)}{Gbest}$. '*Gbest*' is the best value collected by all the compared algorithms. As seen from the data in Table 9, the proposed IEHO is obviously superior to other algorithms. In Table 10, *t*-test (*A,B*) represents the paired *t*-test between algorithm *A* and algorithm *B*. If the *p*-value is less than 0.05, *A* performs better than *B*. As observed from Table 10, all the *p*-values are smaller than 0.05. This indicates that the performance of our IEHO is superior to that of other algorithms, which is consistent with the previous analyses.

**Table 9.** Comparison results of the values of RPDs.

| Instance | Gbest | IEHO | | IEHORR | | IEHONL | | GA | |
|---|---|---|---|---|---|---|---|---|---|
| | | BRPD | APRD | BRPD | APRD | BRPD | APRD | BRPD | APRD |
| RM01 | 38,929 | 0.00 | 0.63 | 13.16 | 15.11 | 5.54 | 6.85 | 26.72 | 29.62 |
| RM02 | 40,938 | 0.00 | 0.56 | 11.23 | 13.55 | 6.31 | 7.72 | 29.04 | 32.42 |
| RM03 | 39,021 | 0.00 | 1.12 | 12.19 | 14.81 | 8.95 | 9.96 | 34.66 | 39.09 |
| RM04 | 37,085 | 0.00 | 1.32 | 13.06 | 15.80 | 11.55 | 14.76 | 43.14 | 46.83 |
| RM05 | 48,178 | 0.00 | 0.49 | 8.02 | 9.78 | 2.52 | 4.39 | 20.75 | 23.02 |
| RM06 | 45,182 | 0.00 | 1.19 | 10.57 | 12.61 | 7.20 | 9.20 | 26.86 | 32.45 |
| RM07 | 42,720 | 0.00 | 0.64 | 11.07 | 13.20 | 7.18 | 9.87 | 34.62 | 38.15 |
| RM08 | 43,474 | 0.00 | 1.68 | 9.87 | 12.78 | 6.32 | 10.16 | 37.74 | 41.44 |
| RM09 | 25,671 | 0.00 | 1.28 | 6.08 | 8.11 | 4.72 | 8.79 | 18.02 | 21.93 |
| RM10 | 23,450 | 0.00 | 2.64 | 3.07 | 7.87 | 7.17 | 11.63 | 21.93 | 25.07 |
| RM11 | 23,792 | 0.00 | 1.84 | 4.24 | 7.63 | 6.55 | 11.19 | 25.64 | 28.96 |
| RM12 | 19,690 | 0.00 | 3.01 | 5.78 | 9.88 | 9.40 | 14.06 | 29.34 | 34.63 |
| RM13 | 46,292 | 0.00 | 0.87 | 7.33 | 8.97 | 4.13 | 6.46 | 21.21 | 23.90 |
| RM14 | 44,993 | 0.00 | 0.52 | 8.80 | 11.11 | 5.88 | 8.88 | 27.30 | 31.03 |
| RM15 | 37,352 | 0.00 | 1.60 | 13.46 | 16.22 | 9.50 | 12.85 | 39.66 | 41.60 |
| RM16 | 43,152 | 0.00 | 1.12 | 10.81 | 13.91 | 10.26 | 12.69 | 38.57 | 40.18 |
| RM17 | 38,890 | 0.00 | 0.63 | 7.55 | 11.54 | 5.48 | 8.74 | 21.64 | 24.54 |
| RM18 | 40,335 | 0.00 | 2.21 | 10.08 | 12.98 | 10.83 | 13.38 | 25.21 | 28.78 |
| RM19 | 37,171 | 0.00 | 2.29 | 13.13 | 15.99 | 16.42 | 17.64 | 34.58 | 37.02 |
| RM20 | 35,477 | 0.00 | 1.60 | 8.64 | 12.86 | 13.19 | 16.39 | 37.08 | 40.73 |
| RM21 | 36,173 | 0.00 | 0.65 | 9.39 | 11.11 | 6.95 | 8.51 | 22.53 | 25.79 |
| RM22 | 36,315 | 0.00 | 1.53 | 10.54 | 13.56 | 8.57 | 11.88 | 27.61 | 33.02 |
| RM23 | 36,598 | 0.00 | 0.86 | 6.54 | 9.78 | 8.43 | 11.43 | 28.64 | 31.67 |
| RM24 | 33,225 | 0.00 | 1.54 | 9.41 | 11.57 | 11.20 | 14.04 | 33.63 | 37.95 |
| RM25 | 38,923 | 0.00 | 0.40 | 8.41 | 10.79 | 3.19 | 5.04 | 20.09 | 23.27 |
| RM26 | 36,294 | 0.00 | 1.16 | 9.88 | 12.11 | 6.16 | 8.72 | 27.91 | 31.32 |
| RM27 | 33,863 | 0.00 | 1.15 | 10.16 | 13.38 | 9.19 | 12.14 | 29.94 | 34.57 |
| RM28 | 31,620 | 0.00 | 1.98 | 9.38 | 13.32 | 8.92 | 12.50 | 34.59 | 41.08 |
| RM29 | 32,118 | 0.00 | 1.09 | 7.64 | 10.37 | 7.91 | 10.41 | 21.72 | 25.50 |
| RM30 | 32,523 | 0.00 | 0.95 | 10.10 | 12.56 | 12.15 | 14.99 | 30.50 | 32.54 |
| RM31 | 32,485 | 0.00 | 1.17 | 8.16 | 10.05 | 12.00 | 13.51 | 28.59 | 33.05 |
| RM32 | 31,072 | 0.00 | 2.17 | 7.59 | 11.04 | 11.46 | 13.99 | 35.46 | 41.15 |
| Mean | - | 0.00 | 1.31 | 9.23 | 12.01 | 8.29 | 11.02 | 29.22 | 33.00 |

| Instance | Gbest | PSO-GA | | VNS | | DABO | | CosSDCSO | |
|---|---|---|---|---|---|---|---|---|---|
| | | BRPD | APRD | BRPD | APRD | BRPD | APRD | BRPD | APRD |
| RM01 | 38,929 | 5.81 | 8.19 | 10.75 | 13.53 | 6.90 | 9.41 | 8.43 | 9.76 |
| RM02 | 40,938 | 7.35 | 9.66 | 13.43 | 16.77 | 5.50 | 8.06 | 11.44 | 12.02 |
| RM03 | 39,021 | 10.03 | 12.64 | 16.13 | 18.81 | 10.68 | 12.60 | 14.87 | 17.42 |
| RM04 | 37,085 | 17.57 | 20.48 | 22.62 | 29.02 | 11.40 | 15.10 | 23.79 | 26.30 |

**Table 9.** *Cont.*

| Instance | Gbest | PSO-GA | | VNS | | DABO | | CosSDCSO | |
|---|---|---|---|---|---|---|---|---|---|
| | | *BRPD* | *APRD* | *BRPD* | *APRD* | *BRPD* | *APRD* | *BRPD* | *APRD* |
| RM05 | 48,178 | 2.24 | 4.78 | 5.69 | 8.26 | 0.33 | 4.39 | 4.46 | 5.23 |
| RM06 | 45,182 | 6.32 | 10.09 | 12.01 | 14.63 | 4.98 | 6.55 | 8.63 | 9.99 |
| RM07 | 42,720 | 9.40 | 12.24 | 14.77 | 17.82 | 3.80 | 4.92 | 9.40 | 10.65 |
| RM08 | 43,474 | 10.64 | 12.78 | 15.90 | 17.54 | 4.24 | 6.16 | 12.57 | 13.42 |
| RM09 | 25,671 | 7.39 | 8.59 | 8.21 | 13.29 | 4.30 | 6.37 | 8.42 | 9.59 |
| RM10 | 23,450 | 6.52 | 9.89 | 13.86 | 18.25 | 4.73 | 7.20 | 16.09 | 17.70 |
| RM11 | 23,792 | 9.91 | 13.54 | 15.30 | 18.87 | 3.14 | 6.62 | 17.41 | 19.60 |
| RM12 | 19,690 | 9.44 | 13.08 | 20.99 | 24.24 | 5.69 | 8.88 | 25.33 | 27.09 |
| RM13 | 46,292 | 4.39 | 8.92 | 3.23 | 8.72 | 4.39 | 6.47 | 4.74 | 5.73 |
| RM14 | 44,993 | 6.65 | 10.64 | 9.18 | 11.19 | 2.13 | 4.47 | 7.91 | 8.17 |
| RM15 | 37,352 | 10.27 | 14.15 | 17.96 | 20.60 | 6.94 | 8.14 | 13.09 | 15.12 |
| RM16 | 43,152 | 13.19 | 15.52 | 19.15 | 21.10 | 4.21 | 6.15 | 13.62 | 15.16 |
| RM17 | 38,890 | 4.69 | 7.60 | 6.71 | 9.62 | 0.57 | 2.51 | 3.07 | 4.60 |
| RM18 | 40,335 | 10.80 | 12.20 | 11.37 | 13.36 | 2.79 | 4.50 | 8.25 | 9.70 |
| RM19 | 37,171 | 16.74 | 18.90 | 19.54 | 22.51 | 6.24 | 9.03 | 17.20 | 18.58 |
| RM20 | 35,477 | 14.19 | 17.70 | 19.05 | 22.76 | 6.25 | 7.88 | 13.64 | 15.22 |
| RM21 | 36,173 | 5.82 | 8.84 | 8.53 | 11.61 | 5.53 | 6.67 | 7.31 | 8.64 |
| RM22 | 36,315 | 11.82 | 14.32 | 15.05 | 18.99 | 4.89 | 7.19 | 9.68 | 13.09 |
| RM23 | 36,598 | 11.80 | 14.88 | 13.22 | 17.82 | 3.32 | 5.33 | 13.19 | 14.20 |
| RM24 | 33,225 | 13.91 | 19.89 | 20.68 | 23.07 | 6.06 | 8.54 | 17.74 | 19.22 |
| RM25 | 38,923 | 4.27 | 5.91 | 8.85 | 11.08 | 4.26 | 7.58 | 6.98 | 7.93 |
| RM26 | 36,294 | 6.27 | 8.80 | 10.39 | 15.54 | 5.70 | 8.14 | 11.14 | 12.32 |
| RM27 | 33,863 | 10.21 | 15.69 | 19.71 | 22.96 | 6.80 | 9.72 | 17.71 | 19.45 |
| RM28 | 31,620 | 11.44 | 16.73 | 16.57 | 23.72 | 5.56 | 8.52 | 20.23 | 22.86 |
| RM29 | 32,118 | 7.05 | 9.15 | 11.00 | 13.86 | 4.64 | 6.34 | 7.96 | 9.03 |
| RM30 | 32,523 | 11.37 | 15.73 | 15.16 | 18.78 | 7.47 | 8.88 | 14.52 | 16.18 |
| RM31 | 32,485 | 11.18 | 15.88 | 17.19 | 21.20 | 3.85 | 5.80 | 15.32 | 17.34 |
| RM32 | 31,072 | 13.44 | 19.75 | 20.64 | 24.05 | 5.18 | 7.45 | 16.50 | 18.60 |
| Mean | - | 9.44 | 12.72 | 14.15 | 17.61 | 5.08 | 7.36 | 12.65 | 14.06 |

**Table 10.** The *t*-test results of paired samples.

| *t*-Test | *p*-Value (*BRPD*) | *p*-Value (*ARPD*) |
|---|---|---|
| *t*-test(IEHO, IEHORR) | $9.53407 \times 10^{-20}$ | $2.82387 \times 10^{-21}$ |
| *t*-test(IEHO, IEHONL) | $6.25182 \times 10^{-16}$ | $4.34201 \times 10^{-19}$ |
| *t*-test(IEHO, GA) | $2.44515 \times 10^{-22}$ | $2.50859 \times 10^{-23}$ |
| *t*-test(IEHO, PSO-GA) | $2.47955 \times 10^{-15}$ | $6.91352 \times 10^{-17}$ |
| *t*-test(IEHO, VNS) | $1.3474 \times 10^{-16}$ | $1.10045 \times 10^{-18}$ |
| *t*-test(IEHO, DABO) | $1.0809 \times 10^{-13}$ | $2.06793 \times 10^{-15}$ |
| *t*-test(IEHO, CosSDCSO) | $3.34591 \times 10^{-14}$ | $1.5326 \times 10^{-14}$ |

## 6. Conclusions and Future Work

Energy-saving scheduling has attracted more and more attention in recent years and has become a hotspot in the manufacturing area. In this paper, an energy-saving assembly job shop scheduling problem with transportation times is investigated in a manner that is close to actual production. A mathematical model is established with the criteria to minimize the total energy consumption of the workshop. An improved elephant herding optimization algorithm, named IEHO, is developed according to the characteristics of the problem. A number of experiments are conducted to test the performance of the IEHO algorithm. The comparison results demonstrate that IEHO is very competitive in solving the energy-saving assembly job shop scheduling problem with transportation times.

In this paper, only single-objective static scheduling is investigated, thereby restricting the implementation of IEHO for multi-objective dynamic scheduling problems. In the next

work, some more practical factors will be considered, such as multi-objective optimization, dynamic/uncertain manufacturing environments, transportation constraints, worker flexibility, deterioration/learning effect, time-of-use electricity policy, and renewable resources. Moreover, the left-shift decoding method in IEHO concentrates more on the improvement in the solution quality, which leads to an increase in the computational time. Therefore, we will further develop more efficient search strategies, improve the computational efficiency of the algorithm, and implement an effective combination of EHO with other algorithms.

**Author Contributions:** Conceptualization, T.J. and L.L.; methodology, T.J. and L.L.; software, H.Z. and Y.L.; writing—original draft preparation, T.J. and L.L.; writing—review and editing, T.J. and L.L.; funding acquisition, T.J., L.L., H.Z. and Y.L. All authors have read and agreed to the published version of the manuscript.

**Funding:** This research was supported by the Fundamental Research Funds for the Central Universities, JLU; the Natural Science Foundation of Shandong Province (ZR2021MG008, ZR2020QG005, ZR2020QG023); the Youth Entrepreneurship and Technology of Colleges and Universities in Shandong Province (2019KJN002); the Yantai Science and Technology Planning Project (2021xdhz072), the Major Innovation Projects in Shandong Province (2020CXGC010702, 2021CXGC010702), Yantai next generation industrial robot and Intelligent Manufacturing Engineering Laboratory.

**Institutional Review Board Statement:** Not applicable.

**Informed Consent Statement:** Not applicable.

**Data Availability Statement:** The data used to support the findings of this study are included within the article.

**Conflicts of Interest:** The authors declare no conflict of interest.

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
