# Peer review of "An Improved Elephant Herding Optimization for Energy-Saving Assembly Job Shop Scheduling Problem with Transportation Times"

_axioms, doi:10.3390/axioms11100561_

Round 1
Reviewer 1 Report
I found the manuscript to be well written. The abstract is a bit cumbersome but this can be fixed
Two minor corrections are required
First in "(5) Each machine can be turned off until all jobs on it are completed." is there a typo.e.g. is there a missing "not"
Second, on p.4 you write "Gamma : A big positive number;". I see it enters constraints but please explain this choice of variable to the reader.
Author Response
- The reviewer’s comment:
I found the manuscript to be well written. The abstract is a bit cumbersome but this can be fixed.
The author’s response:
Following the comment, we have revised the abstract. Please see Page 1.
- The reviewer’s comment:
Two minor corrections are required. First in "(5) Each machine can be turned off until all jobs on it are completed." is there a typo.e.g. is there a missing "not"; Second, on p.4 you write "Gamma : A big positive number;". I see it enters constraints but please explain this choice of variable to the reader.
The author’s response:
Following this comment, some revisions have been conducted as below.
Firstly, we have revised the sentence. Please see Page 4.
Secondly, Gamma is set to be big enough to ensure that Constraints (8) and (9) holds.we have explain the Gamma in Constraints (8) and (9). Please see Pages 5 and 6.

Reviewer 2 Report
Dear Authors,
The manuscript “An improved Elephant Herding Optimization for Energy-Saving Assemble Job Shop Scheduling Problem with Transportation Times” is presented. An improved algorithm is proposed for the features of the standard Elephant Herding Optimization Algorithm. The main improvement is incorporating an encoding approach to represent the scheduling solution and an energy-saving decoding strategy. The manuscript is well organized and redacted. The problem description and mathematical model are well formulated. All numerical experiments are supported by statistical analysis.
Finally, In my opinion, I suggest that this article should be accepted after minor revision.
Therefore, I suggest the following corrections:
1. The time complexity (also known as computational complexity) of the proposed algorithm should be included.
e.g., See section 2.3.1 in https://www.mdpi.com/2227-7390/10/1/102/htm
2. A convergence graph for each instance must be plotted and include all the algorithms used in the experimentation.
On the other hand, I am curious about the use of a virtual machine with an obsolete operating system and so little ram memory.
Author Response
- The reviewer’s comment:
The time complexity (also known as computational complexity) of the proposed algorithm should be included.
The author’s response:
According to the comment, we have added the analysis of time complexity of the algorithm. Please see Pages 12-13.
- The reviewer’s comment:
A convergence graph for each instance must be plotted and include all the algorithms used in the experimentation.
The author’s response:
Following the comment, we have added the convergence curves of algorithms on 12 randomly selected instances. Please see Figure 8 on Page 22.
- The reviewer’s comment:
On the other hand, I am curious about the use of a virtual machine with an obsolete operating system and so little ram memory.
The author’s response:
We have rechecked the virtual machine. The RAM memory was set to be 6G.
